materials science/biochemistry/biomaterials

poly (2-hydroxyethyl acrylate), hydroxyl oxidation, aldehyde group, collagen, cross-linking, modification

**Author for correspondence:**
Baohua Liu
e-mail: 351551804@qq.com

This article has been edited by the Royal Society of Chemistry, including the commissioning, peer review process and editorial aspects up to the point of acceptance.

# Structure validation of oxidized poly (2-hydroxyethyl acrylate) with multiple aldehyde groups and its application for collagen modification

Baohua Liu, Jian Wang, Lili Ji, Ting Bai, Yin Zhang and Dayu Liu

College of Food and Biological Engineering, Sichuan Key Laboratory of Meat Processing, Chengdu University, Chengdu, Sichuan 610106, People's Republic of China

BL, 0000-0001-8211-8165

The structural characteristic of oxidized poly (2-hydroxyethyl acrylate) (OP) was confirmed by high-performance liquid chromatography, gel permeation chromatography and hydroxylamine hydrochloride titration. The results demonstrated that OP prepared through 2,2,6,6-tetramethylpiperidine-1-oxyl-mediated oxidation of poly (2-hydroxyethyl acrylate) was featured by multiple aldehyde groups on its side chain, with no free formaldehyde produced during the oxidation process. The computational simulation for the electrophilic reactivity of OP molecule showed that the reactivity of the aldehyde groups in OP with the amino groups of collagen was comparable to that of glutaraldehyde. In this study, OP was chosen as a collagen modifier to investigate the modification effects on the secondary structure, aggregation behaviour and thermal stability of collagen. The covalent cross-linking occurred between the aldehyde groups of OP and the amino groups of collagen under alkaline condition. The covalent binding between OP and collagen was strengthened with the increasing reaction pH and OP dosage, and the triple helix of collagen was altered to some degree. Furthermore, OP promoted the intense aggregation of collagen and enhanced the thermal stability of collagen. This work provides guidance for preparing novel collagen modifier with multiple aldehyde groups.

# 1. Introduction

Poly (2-hydroxyethyl acrylate) (PHEA) is the homopolymer of water-soluble acrylic monomer 2-hydroxyethyl acrylate (HEA), possessing a large number of primary hydroxyls on the side chains. Due to the superior thermal stability, hydrophilic and biocompatibility [1–3], PHEA has inspired the development of biomedical materials, such as cartilage implantation interpenetrating network accessories, drug sustained-release micelles and porous hydrogels [4–6]. In particular, PHEA could be further oxidized partially through the selective 2,2,6,6-tetramethylpiperidine-1-oxyl (TEMPO)-mediated oxidation of its primary hydroxyl groups, thus introducing multiple functional aldehyde groups on the molecular chain of the oxidized poly (2-hydroxyethyl acrylate) (OP) [7]. Based on the structure of multiple aldehyde groups, OP was used as a gelatin cross-linker and exhibited good cross-linking reactivity. This fact suggests the reactivity of OP with the amino groups of active protein or polypeptides, which lay a foundation for OP as an effective collagen modifier.

Collagen, the most abundant and widely distributed functional protein in mammals [8,9], has received more attention in the fields of biomedical, such as tissue engineering, aesthetic healthcare and food industry [10–13] due to its superior biocompatibility, biodegradability and low antigenicity [14,15]. However, the special three-helix structure of collagen is easy to be damaged by heating, acids and bases. So collagen suffers from low thermal stability and weak mechanical strength. It is therefore necessary to undertake chemical modification for collagen to meet the application requirements in different fields. Thus, the selection and use of suitable cross-linkers is a key factor for collagen modification. Glutaraldehyde and oxidized polysaccharide, such as dialdehyde starch [16–18], dialdehyde carboxymethyl cellulose [19,20], oxidized sodium alginate [21] and chitosan dialdehyde [22], have been used as protein cross-linkers because of the reactivity of aldehyde group with amine group through the formation of covalent bond (amide and imine linkages). However, one of the drawbacks of glutaraldehyde as a cross-linker or stabilizer was the instability of the Schiff's base [23]. Compared with other oxidized polysaccharide cross-linkers, OP with multiple aldehyde groups is supposed to have better biocompatibility and biodegradability. As summarized above, OP had good cross-linking reactivity with gelatin, so we have sound reason to believe that OP may have the potential to covalently cross-link with collagen to improve its thermal stability (the proposed schematic diagram of the cross-linking reaction is shown in figure 1). This prospect is closely associated with the structural characteristic of multiple aldehyde groups of OP. The structure of OP was preliminarily characterized by Fourier transform infrared (FT-IR) spectroscopy and $^{13}$C NMR in our previous work [7], yet the structure–property relations of OP still needs to be further investigated, which is meaningful for developing macromolecular cross-linkers with multiple aldehyde groups for collagen.

In the present work, the structure and properties of OP samples, including aldehyde content, relative molecular weight and the free formaldehyde content, were further investigated by high-performance liquid chromatography (HPLC), gel permeation chromatography (GPC) and hydroxylamine hydrochloride titration. Furthermore, the electrophilic reactivity of OP was confirmed by computational simulation. Then, the modification effects of OP on collagen were evaluated by using FT-IR, circular dichroism (CD), atomic force microscopy (AFM), microthermal differential scanning calorimeter (VP-DSC) and thermogravimetric (TG) analyses.

# 2. Material and methods

## 2.1. Materials

2-hydroxyethyl acrylate (HEA, 96%, purity) and porcine type I collagen purchased from Sigma-Aldrich LLC (USA). Acrylic acid (AA) and other chemicals were of analytical grade and purchased from Chengdu Kelong Chemical Co., Ltd (China).

## 2.2. Preparation of oxidized poly (2-hydroxyethyl acrylate)

A series of OP labelled as OP10, OP30, OP60, OP100 and OP140 (the schematic diagram of OP structure is shown in figure 1) were prepared according to the previous experimental procedure [7]. In brief, first, PHEA was synthesized via aqueous solution polymerization of HEA. Then, 0.055 g TEMPO and 2.67 g NaBr were added into 86.8 g PHEA solution (containing approx. 0.26 mol hydroxyl groups) at ambient temperature under stirring, pH of the mixed solution was adjusted to 9.4 using 1.0 mol l$^{-1}$ NaOH

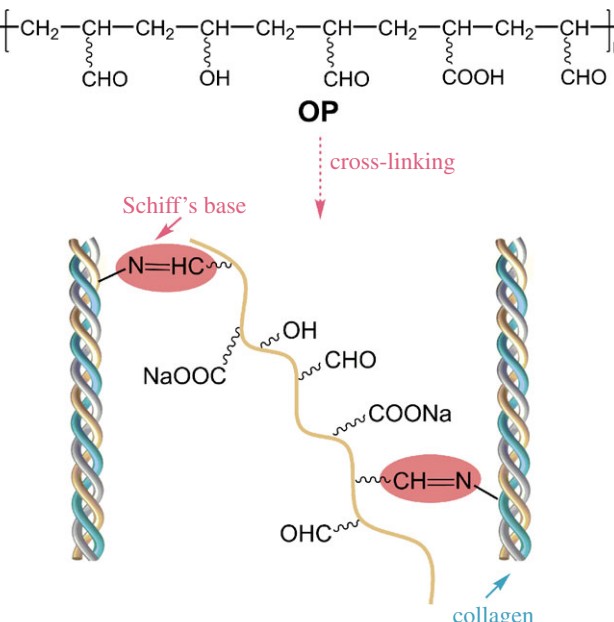

**Figure 1.** Schematic diagram of OP structure and the proposed cross-linking reaction with collagen.

solution. Subsequently, 9.2–128.8 g NaClO solution (accounting for 10%, 30%, 60%, 100% and 140% of the hydroxyl groups in HEA, respectively), with the pH also adjusted to 9.4, was dropwise added into the PHEA mixture to initiate the oxidation reaction. The reaction temperature was set as 5°C, and $1.0 \, \text{mol} \, l^{-1}$ NaOH solution was also dropwise added to maintain the set pH during the reaction. The oxidation reaction duration was 90 min and then stopped by adding 5 ml 95% (v/v) ethanol. Furthermore, HEA was oxidized according to the procedure as OP30. The oxidation product was labelled as OHEA, which was used as a control group of OP products.

## 2.3. Structure validation of oxidized poly (2-hydroxyethyl acrylate)

In this paper, the structure–property relations of OP were further explored by HPLC, GPC and hydroxylamine hydrochloride titration. Then, the computational simulation was used to investigate the electrophilic reactivity of OP, in an attempt to provide direct and quantitative evidence for the cross-linking reaction between OP and collagen.

## 2.4. Reaction between oxidized poly (2-hydroxyethyl acrylate) and collagen

Collagen solution ($0.5 \, \text{mg ml}^{-1}$) was prepared by dissolving 0.05 g lyophilized collagen into 100 ml $0.5 \, \text{mol} \, l^{-1}$ acetic acid solution under stirring at 25°C for 24 h. Subsequently, 5% OP (w/w, based on dry collagen mass, the same below) was incorporated under stirring, the reaction pH of the mixture was adjusted with $1.0 \, \text{mol} \, l^{-1}$ NaOH solution to 4, 6, 8 and 10, respectively. Then the reaction was conducted at 25°C for 24 h. The other three trials were conducted under the same conditions except that different dosages of OP (0%, 1% and 2%, respectively) were used, and the pH was fixed at 8.0. The OP modified collagen products (labelled as OPC) were collected for further characterization and analysis. Besides, glutaraldehyde modified collagen (GC) was prepared as a comparison sample according to the procedure above except that glutaraldehyde dosage was 0.02%, and pH was set as 8.0. Due to the fact that glutaraldehyde exhibited strong cross-linking effect with collagen under alkaline condition, the dosage of glutaraldehyde was much less than OP to avoid the precipitation of collagen and thus hinder the following analysis and comparison of the modified collagen.

## 2.5. Characterization and analysis

### 2.5.1 High-performance liquid chromatography

To further verify the structure of OP, the samples of HEA and OHEA were recorded using HPLC (Agilent LC-1100, Agilent Technologies, USA) with Aichrombond-AQ C18 column (250 mm × 4.6 mm, 5 µm) and

DAD ultraviolet detector, the detection wavelength was 210 nm. Ten millilitres test sample (25 µg ml$^{-1}$) was filtered through a 0.22 µm pore membrane at first. The injection volume was 10 µl. The mobile phase A was 0.35% (w/w) phosphoric acid (pH = 2), and the mobile phase B was chromatographic methanol with a flow rate of 0.8 ml min$^{-1}$ under 40°C. Acrylic acid (AA) was used as a comparison sample for the HPLC analysis.

### 2.5.2. Determination of free formaldehyde content of oxidized poly (2-hydroxyethyl acrylate)

The free formaldehyde content of OP (prepared under the optimal condition) was measured mainly on the basis of the standard method (ISO 27587-2009) [24] with some modification [25]. In brief, 0.2 ml OP sample was accurately incubated into U-tube at 90°C oil-bath. The released formaldehyde was captured by 25 ml 2, 4-dinitrophenylhydrazine absorption solution accompanied by nitrogen gas purging for 30 min. Then, the absorption solution was diluted to 50 ml and detected using HPLC (Agilent LC-1100, Agilent Technologies, USA) with Aichrombond-AQ C18 column (250 mm × 4.6 mm, 5 µm) and DAD ultraviolet detector; the detection wavelength was 360 nm. The mobile phase A was deionized water, and the mobile phase B was chromatographic acetonitrile with a flow rate of 0.8 ml min$^{-1}$ under 30°C. Formaldehyde standard solution (1 mg l$^{-1}$) was used as a control for analysis. In addition, a series of formaldehyde standard solutions (1 mg l$^{-1}$, 2 mg l$^{-1}$, 4 mg l$^{-1}$, 6 mg l$^{-1}$, 8 mg l$^{-1}$ and 10 mg l$^{-1}$) were prepared and determined by HPLC under the conditions above for the formaldehyde standard curve plotting.

### 2.5.3. Molecular weight determination

The molecular weight of OP was determined by using GPC (Viscotek 270 max, Malvern, UK) equipped with TSK-gel GMPWXL column (7.8 × 300 mm, Tosoh, Japan). Ten millilitres OP aqueous solution (5 mg ml$^{-1}$) was filtered through a 0.22 µm pore membrane at first. The injection volume was 100 µl. The eluent was 0.1 mol l$^{-1}$ NaNO$_3$ at a flow rate of 0.6 ml min$^{-1}$ under 25°C. Polyethylene oxide standard (2 mg ml$^{-1}$, $M_w = 2.7 \times 10^4$; Tosoh, Japan) was used for calibrated before measurement. Weight-average molecular weights ($M_w$), number-average molecular weights ($M_n$) and polydispersities ($M_w/M_n$) of OP samples were calculated with a dn/dc value of 0.116 ml g$^{-1}$.

### 2.5.4. Determination of aldehyde content

Aldehyde contents of OP samples were determined by using hydroxylamine hydrochloride titration as described in our previous work [7]. Measurements were made in triplicate. Results are shown as mean ± standard deviation.

### 2.5.5. Computational simulation for the electrophilic reactivity of oxidized poly (2-hydroxyethyl acrylate)

The reaction between aldehyde groups of OP and amine groups of collagen is a kind of nucleophilic addition, referring to as the lone pair electrons on the nitrogen atom of amino group attack the carbon atom of aldehyde group. In general, the stronger electro-positivity of aldehyde carbon in OP signifies the higher electrophilic reactivity of OP molecule. To explore the electrophilic reactivity of OP molecule, OP model molecules with a polymerization degree of three were constructed (the structural formulae of OP model molecules are shown in figure 2). The optimized equilibrium structures of OP model molecules were obtained based on density functional theory with B3LYP hybrid functional [26]. Gaussian 09 computational package was used for the calculation of the charge of aldehyde carbon, and 6–31G$^{**}$ basis set was chosen [27]. Meanwhile, glutaraldehyde was used as a control for analysis of the electrophilic reactivity.

### 2.5.6. Fourier transform infrared spectroscopy

The FT-IR spectra of lyophilized collagen, OP and modified collagen (OPC at different pH values and GC) were recorded in the wavenumber ranging from 500 to 4000 cm$^{-1}$ using FT-IR spectrometer (Nicolet IS 10, Thermo Scientific, USA) with a resolution of 4 cm$^{-1}$. All the samples were analysed using KBr disc method.

**Figure 2.** Structural formulae of OP model molecules and glutaraldehyde: (*a*) OP-A containing three aldehyde groups, (*b*) OP-B containing two aldehyde groups, (*c*) OP-C containing one aldehyde group and (*d*) glutaraldehyde.

### 2.5.7. Circular dichroism measurement

The secondary conformation of collagen samples (collagen, 0.5 mg ml$^{-1}$; OPC at different pH values and GC) were measured in the far UV region (190–250 nm) using CD spectropolarimeter (J-810, Jasco, Japan) under nitrogen atmosphere at 25°C, with an average of three scans at a speed of 100 nm min$^{-1}$. Acetic acid (0.5 mol l$^{-1}$) was recorded as a reference. The resulting spectra were expressed in terms of molar ellipticity [28].

### 2.5.8. Atomic force microscopy analysis

Collagen, OPC with different OP dosages and GC solutions were diluted to 5 mg l$^{-1}$. Then, 5 µl of the diluted solutions were spread on the newly peeled mica substrate and kept for drying overnight. The morphologies of collagen samples were observed using AFM (SPM-9600, SHIMADZU, Japan) in tapping mode.

### 2.5.9. Microthermal differential scanning calorimeter analysis

DSC thermograms of collagen samples (collagen, 0.5 mg ml$^{-1}$; OPC at different pH values, OPC with different OP dosages and GC) were recorded using VP-DSC (MicroCal, USA) over a temperature range of 20 to 70°C, at a heating rate of 1.5°C min$^{-1}$ under nitrogen atmosphere. The endothermal peak temperature was denoted as denaturation temperature ($T_d$) of collagen.

### 2.5.10. Thermogravimetric analysis

Thermal decomposition analysis of lyophilized OP and collagen samples (collagen, OPC with different OP dosages and GC) was carried out using Thermal Gravimetric Analyzer (Pyris I, PerkinElmer, USA). About 5 mg specimen per sample was taken for the analysis from 25 to 600°C at a heating rate of 10°C min$^{-1}$ under nitrogen atmosphere, and the flow rate of nitrogen was 20 cm$^3$ min$^{-1}$.

# 3. Results and discussion

## 3.1. Structure validation of oxidized poly (2-hydroxyethyl acrylate)

### 3.1.1. High-performance liquid chromatography

OP with multiple aldehyde groups was prepared by using selective TEMPO-mediated oxidation of PHEA under alkaline condition. The structural characteristic of multiple aldehyde groups of OP was preliminarily confirmed by FT-IR and $^{13}$C NMR analyses in our former research [7]. In theory, the ester group of PHEA was possibly hydrolysed to generate polyacrylic acid (PAA) molecular chain segment under strong alkaline condition during the oxidation process of PHEA. In that case, only a small amount of primary hydroxyl groups would be selectively oxidized into aldehyde groups, the likely side reaction is shown in figure 3. In order to verify this speculation, HEA was selected as a model, and OHEA as the model of OP was analysed by HPLC. AA was chosen as the control group for analysis. HPLC spectra of the samples are shown in figure 4. The HPLC spectrum of HEA depicted a single peak with the retain time at 13.540 min, and the retain time of OHEA was 13.548 min accompanied by a very small peak at 9.731 min. By comparison, there was a prominent single peak at 9.718 min for AA. Although the retain times of OHEA and HEA were close to each other, the peak areas of the two samples were different from each other, 594 005 for OHEA, while 809 106 for HEA (electronic supplementary material, table S1). It was probably because the structure of OHEA was similar to HEA; only the functional groups on side chains of the two samples were different. Whereas the retain time of the main peak in OHEA was quite different from AA, and the area (21 103) of the main peak at 9.731 min was far less than that of AA (867 403), so the amount of trace AA in OHEA was calculated as only 2.43% based on the peak area. Therefore, the HPLC analysis refuted this speculation and verified that the side reaction barely happened during the selective oxidation of PHEA, and it also indirectly confirmed the structural characteristic of multiple aldehyde groups of OP.

### 3.1.2. Free formaldehyde content

Free formaldehyde content of OP was detected by HPLC, as shown in figure 5b. As for the control group, formaldehyde reacted with 2, 4-dinitrophenylhydrazine to generate phenylhydrazone, displaying a distinct unimodal peak at 6.34 min. By contrast, OP exhibited a tiny signal peak with the peak area of 116.79, which suggested that only a trace amount of formaldehyde existed in OP30. The content of formaldehyde was calculated as merely 30.85 mg l$^{-1}$ according to the standard curve of formaldehyde solution (figure 5a).

Thus, it can be seen that there was very little chance that the primary carbon atoms on the side chains of PHEA ruptured to produce free formaldehyde during oxidation. In other words, all the aldehyde groups of OP were almost originated from the selective TEMPO-mediated oxidation of hydroxyl groups of PHEA. The result was consistent with the HPLC analysis and further confirmed the structural characteristic of multiple aldehyde groups of OP.

The aldehyde content of OP is shown in electronic supplementary material, table S2; it increased first and then decreased with increasing of the oxidation degree from OP10 to OP140, and OP30 owned the highest aldehyde content (3.66 mmol g$^{-1}$). Meanwhile, the weight-average molecular weight ($M_w$) of OP varied little as a whole (20 991 ∼ 24 811 g mol$^{-1}$) compared with PHEA (21 638 g mol$^{-1}$), the number-average molecular weight ($M_n$) was decreased from 8522 to 2242 g mol$^{-1}$, the polydispersity index ($M_w/M_n$) became wider in the range of 2.463 to 10.479. These facts further indicated that PHEA molecular chain was not randomly broken into fragments during the selective TEMPO-mediated oxidation. The aldehyde content depends on the oxidation degree of OP. The higher degree of oxidation was used, the lower content of aldehyde group of OP was obtained. This should be ascribed to the fact that a part of aldehyde groups was further oxidized into carboxyl groups by surplus NaClO. Therefore, the moderate controlling of the oxidation degree is a critical factor to obtain OP with more aldehyde groups. Based on the results above, OP30 with the highest aldehyde content was selected as a modifier for collagen in the following investigation.

### 3.1.3. Electrophilic reactivity

Based on the structural characteristic of multiple aldehyde groups of OP, in principle, the nucleophilic addition reaction between aldehyde groups of OP and amino groups of collagen could occur to

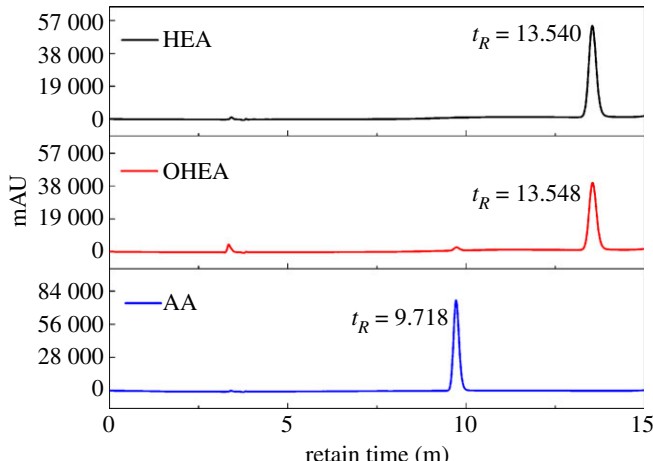

**Figure 3.** Schematic diagram of TEMPO-mediated oxidation of PHEA and the possible side reaction.

**Figure 4.** HPLC spectra of HEA, OHEA and AA.

realize the covalent cross-linking among collagen molecules. In order to provide a direct and quantitative basis for the modification of collagen by OP, it is necessary to evaluate the electrophilic reactivity of OP. Figure 6 illustrates the optimized structures of OP and glutaraldehyde model molecules, the charges of aldehyde carbons of which were used to evaluate their electrophilic reactivity. As shown in electronic supplementary material, table S3, the charge of aldehyde carbon in each OP model molecule (OP-A, OP-B and OP-C) was approximately the same as that of aldehyde carbon in glutaraldehyde, the values were all about 3.6. The result indicated that electrophilic reactivity of the aldehyde groups in OP was comparable to that of glutaraldehyde, which laid a good foundation for the covalent cross-linking between OP and collagen.

## 3.2. Modification of collagen by oxidized poly (2-hydroxyethyl acrylate)

### 3.2.1. Fourier transform infrared analysis

Figure 7a shows the FT-IR spectra of I-collagen, OP, OP-modified collagen (OPC) and glutaraldehyde modified collagen (GC). The spectrum of I-collagen showed that the absorption bands at 3325 and 3080 cm$^{-1}$ were ascribed to amide A (N-H or O-H stretch) and amide B (N-H coupled with C-H stretch) absorption, respectively; the bands located at 1657, 1550 and 1240 cm$^{-1}$ were corresponding to the amide I (C=O stretch), amide II (N-H bend) and amide III (C-H stretch) absorption, respectively. These characteristic peaks demonstrated the special triple-helical conformation of collagen. Compared with the spectrum of OP (figure 7a(g)), the characteristic absorption peak of the aldehyde group at 1730 cm$^{-1}$ did not exist in the FT-IR spectra of OPC. Moreover, it was worthy to note that when the reaction pH was below 8, a new peak at 1640 cm$^{-1}$ corresponding to the imine linkage (−C=N) was formed due to the Schiff's base reaction between aldehyde and amino groups [29], which was similar to the FT-IR spectrum of GC. While when the

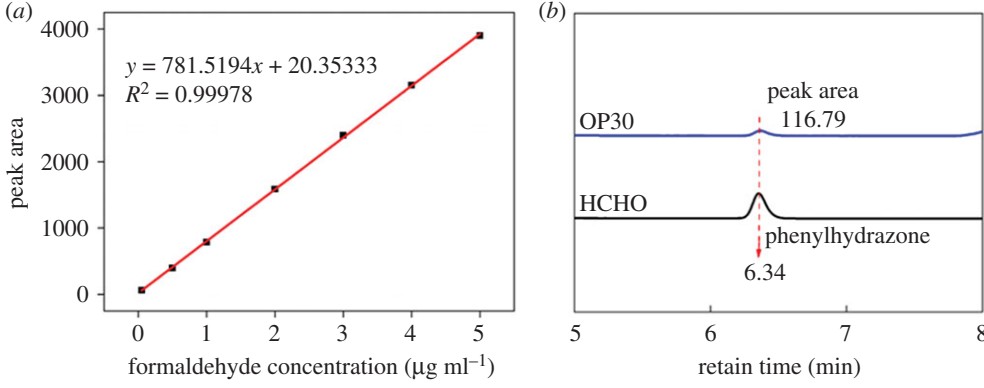

**Figure 5.** Standard curve of formaldehyde solution (*a*) and HPLC spectrum of OP (*b*).

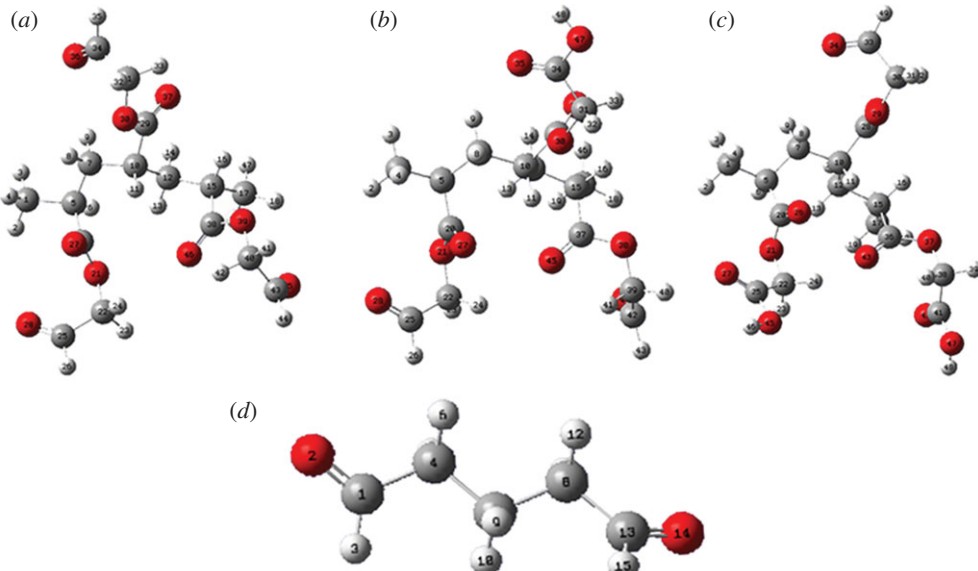

**Figure 6.** Optimized structures of OP and glutaraldehyde model molecules: (*a*) OP-A containing three aldehyde groups, (*b*) OP-B containing two aldehyde groups, (*c*) OP-C containing one aldehyde group and (*d*) glutaraldehyde.

reaction pH was above 6, the imine peak (C=N) was found to be overlapped with the amide I band of collagen, which was possibly ascribed to the strong covalent binding between OP and collagen under alkaline condition. This can also be reflected from the decreased absorption intensity of hemiacetal and hydrated aldehyde at 820–880 cm$^{-1}$ [30,31], especially at pH 8 and 10. In addition, the characteristic absorption peaks of amide II and amide III in OPC also presented slight shift and intensity decreased. All these observations manifested the covalent cross-linking between OP and collagen, and the covalent binding was strengthened with increasing reaction pH. Meanwhile, the triple helix structure of collagen may be changed to some degree due to the covalent cross-linking effect.

### 3.2.2. Circular dichroism spectra

The secondary structure of OPC at different reaction pH was assessed by CD, as shown in figure 7*b*. Just as described in the literature, the CD spectrum of type I collagen exhibited a positive maximum peak at 210–230 nm, and the negative minimum peak at 190–200 nm [32], which demonstrated the special triple-helical conformation of native collagen. The CD spectra of OPC at different pH were basically identical with that of I-collagen and GC, except that the signal intensity of the negative minimum peak and the positive maximum peak decreased continuously with increasing reaction pH. Notably, the positive maximum peak of OPC at pH 10 totally disappeared. It is well known that the disappearance of the positive maximum peak is indicative of the complete denaturation of collagen [33,34]. Here, when the

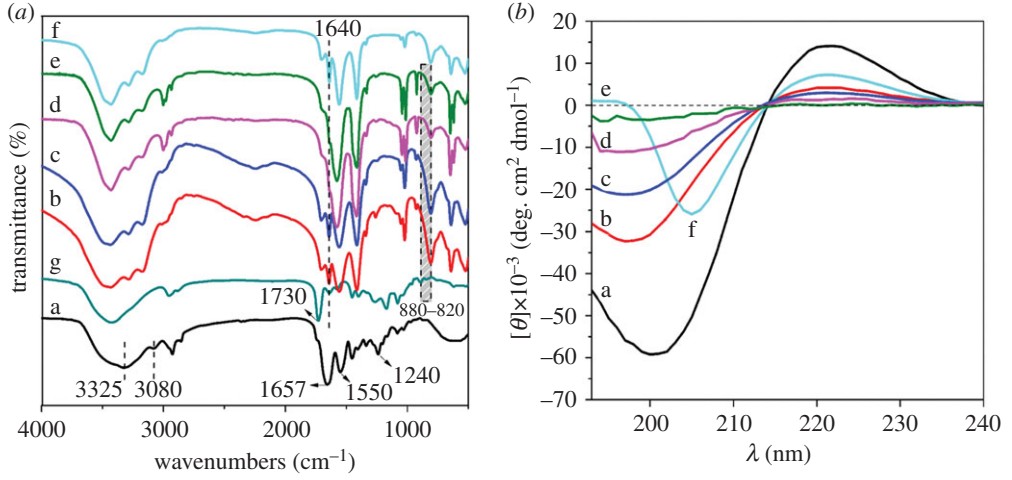

**Figure 7.** FT-IR spectra (*a*) and CD spectra (*b*): I-collagen (a), OPC at pH 4 (b), pH 6 (c), pH 8 (d), pH 10 (e), GC (f) and OP (g).

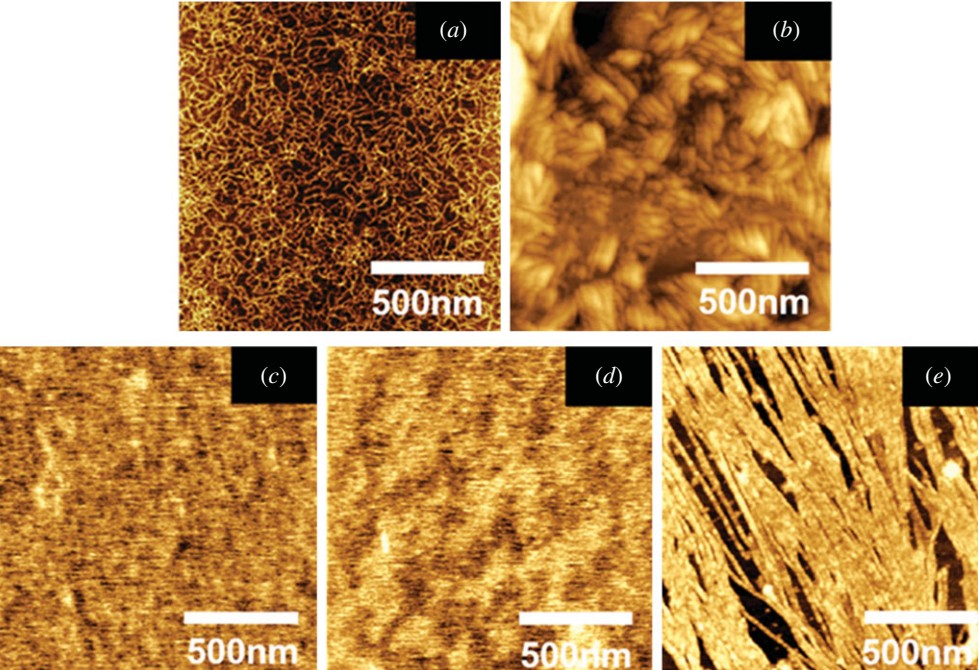

**Figure 8.** AFM images of OPC with different OP dosages: 0% (*a*), 1% (*b*), 2% (*c*), 5% (*d*) and GC (*e*).

reaction pH was above 8, the triple helix structure of collagen might be altered gradually till the complete denaturation, which was attributed to the fact that alkaline condition benefited the generation of Schiff's base between the aldehyde group of OP and the amino group of collagen, thus reinforced the covalent cross-linking effect between OP and collagen. The CD spectra further confirmed the occurrence of the covalent cross-linking between OP and collagen, which was in accordance with the result of FT-IR. Meanwhile, OP probably changed the aggregation state of collagen to a degree under alkaline condition.

### 3.2.3. Atomic force microscopy analysis

The influence of OP on the aggregation morphology of collagen was observed by AFM, as shown in figure 8. I-Collagen randomly self-aggregates into curled microfibres—minuscule hair-like filaments (figure 8*a*). After covalently binding with OP, the aggregation morphology of collagen changed obviously. When the dosage of OP was 1% (figure 8*b*), the OPC assembled into stumpy and compact rod-like bundles of fibres. When the dosage of OP increased to 2%, the aggregation degree of collagen molecules enhanced evidently, showing a dense membranous structure (figure 8*c*). With further

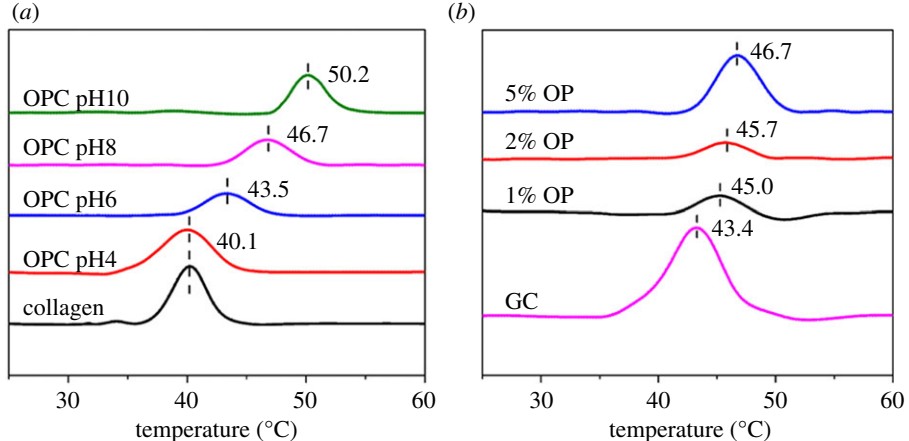

**Figure 9.** VP-DSC thermograms of OPC at different pH values (*a*) and OPC with different OP dosages (*b*).

increase of the dosage of OP to 5% (figure 8*d*), the OPC displayed some degree of uneven agglomeration. Accordingly, the aggregation degree of collagen was strengthened with the increasing of OP dosage, which was ascribed to the fact that more dosage of OP would introduce more aldehyde groups, thereby generating stronger covalent binding with collagen molecules. The aggregation morphology of OPC was different from the distinct and well-organized fibres of GC. It was probably due to the structure of multiple aldehyde groups of OP, which was distinguished from the double-aldehyde structure of glutaraldehyde. Moreover, the thermal stability of collagen is supposed to be improved by the intense aggregation of collagen molecules.

### 3.2.4. Thermal stability

It is generally believed that change of the aggregation state of collagen has a great influence on its thermal properties [35], so the thermal stability of OPC was determined by VP-DSC to evaluate the covalent binding effect between OP and collagen. The endothermic peak value usually corresponds to the denaturation temperature ($T_d$) of collagen that indicates the transformation of triple helical to the random coil. The influence of reaction pH on the $T_d$ of OPC is shown in figure 9*a*. When the reaction pH was 4, $T_d$ of OPC was the same as I-collagen (40.1°C), then it showed an increasing trend as the reaction pH rises. $T_d$ of OPC was increased to 50.2°C when the reaction pH was 10, which was improved by almost 10°C compared with I-collagen. The results demonstrated the enhancement of the thermal stability of collagen by OP with the increasing reaction pH. It was mainly attributed to the fact that alkaline condition was beneficial for the covalent cross-linking between aldehyde groups of OP and amino groups of collagen, so did the formation of covalent bonds intra-/inter-collagen molecules. Figure 9*b* showed that $T_d$ of OPC increased with increasing dosage of OP when the cross-linking reaction proceeded at pH 8, and it was even higher than that of GC, which was closely associated with the strong covalent cross-linking effect between OP and collagen. The results were in accordance with the AFM analysis.

TG and DTG curves of OP30, I-collagen, OPC with different OP dosages and GC are depicted in figure 10. All the samples followed multi-step thermal decomposition. As for OP30 (figure 10*a*), some small molecular components might evaporate in the range of 100–250°C, and the major thermal decomposition was centred at 300~600°C with the maximum decomposition rate around 388°C. The secondary structure of collagen was partially changed accompanied by thermal decomposition of collagen chains during the heating process, the maximum decomposition rate was around 306°C (figure 10*b*). When the dosage of OP was 1% (figure 10*d*), the maximum decomposition rate of OPC was elevated to 494°C, which was comparable to that of GC (493°C) as shown in figure 10*c*. Then the maximum decomposition rate of OPC was further increased with the increasing dosage of OP, reaching as high as 516°C when OP dosage was 5% (figure 10*f*), which was about 210°C higher than that of I-collagen. On the whole, the thermal stability of OPC was superior to that of GC, the reason was that the dosage of OP (1%–5%, w/w) was higher than glutaraldehyde (0.02%, w/w), thus induced stronger covalent binding inter/intra collagen molecules. The results of TG and DTG further verified that OP could improve the thermal stability of collagen effectively.

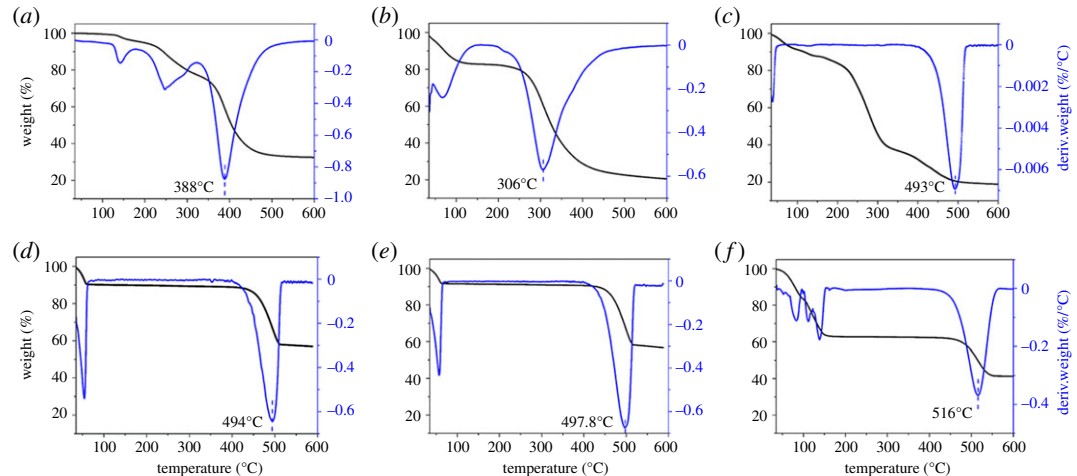

**Figure 10.** TGA and DTG curves of OP30 (*a*), I-collagen (*b*), GC (*c*), OPC with 1%OP (*d*), OPC with 2%OP (*e*) and OPC with 5%OP (*f*) (reaction at pH 8).

# 4. Conclusion

OP is a macromolecular cross-linker with multiple aldehyde groups. The aldehyde groups are originated from the selective TEMPO-mediated oxidation of the hydroxyl groups on the side chain of PHEA, without free formaldehyde produced. The electrophilic reactivity of OP is comparable to glutaraldehyde. OP can generate covalent cross-linking with collagen under the alkaline condition and promote the aggregation of collagen molecules, therefore greatly enhance the thermal stability of collagen. The further understanding of the structure–property relations of OP in this investigation may provide a guide in exploiting macromolecular cross-linker with multiple aldehyde groups for collagen-based materials and promoting the applications of poly (2-hydroxyethyl acrylate) in the fabrication of biomaterials.

Data accessibility. Other data related to my article had already been provided as electronic supplementary material.
Authors' contributions. B.L. and J.W. conceived of the study, performed the experiment and wrote the manuscript; L.J. and T.B. performed the data analyses. Y.Z. and D.L. helped perform the analysis with constructive discussions.
Competing interests. We declare we have no competing interests.
Funding. This work was financially supported by Chengdu Technology Innovation R&D Program (2019YF0501713SN) and Sichuan Key R&D programme (2019YFS0525, 2019YFN0172).
Acknowledgements. The authors especially thank Prof. Zhang Wenhua (Key Laboratory of Leather Chemistry and Engineering (Sichuan University), Ministry of Education, Chengdu, PR China) for the help of computational simulation.

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
