## [Peer Review File · Royal Society Open Science]

Review History

RSOS-201892.R0 (Original submission)

Review form: Reviewer 1

Is the manuscript scientifically sound in its present form?

Yes

Are the interpretations and conclusions justified by the results?

Yes

Is the language acceptable?

No

Do you have any ethical concerns with this paper?

No

Have you any concerns about statistical analyses in this paper?

No

Recommendation?

Accept with minor revision (please list in comments)

Comments to the Author(s)

This manuscript is well organized and written. The authors focused on the covalent cross-linking occurred between the aldehyde groups of OP and the amino groups of collagen under alkaline condition, which sounds very interesting. Here I just wonder that could the author give more evidence on the formation of schiff-base, or could the authors point out the feature of this new chemical bonds via FTIR? Another question, according to my previous study, the concentration of collagen solutions for AFM images in this work were relatively high, I suggest the author do this test at lower concentration.

Review form: Reviewer 2**Is the manuscript scientifically sound in its present form?**

No

Are the interpretations and conclusions justified by the results?

No

Is the language acceptable?

No

Do you have any ethical concerns with this paper?

No

Have you any concerns about statistical analyses in this paper?

No

Recommendation?

Accept as is

Comments to the Author(s)

The manuscript presents a kind of oxidized poly (2-hydroxyethyl acrylate) with multiple aldehyde groups for collagen modification. Though the raw materials the authors used lacks of enough novel content, the idea may be intersted for the collagen researcher. However, the manuscript should rewrite more carefully. It is so hard to understand the manuscript due to irregular writing. The introductions need to pay more attention to the statement on the main innovative points. The analysis should also go deeper. The performances such as mechanical properties, cytotoxicity should be added to make the study more integrated.

Decision letter (RSOS-201892.R0)

This year has been very difficult for everyone, and we want to take the opportunity to thank you for your continued support in 2020.

The Royal Society Open Science editorial office will be closed from the evening of Friday 18 December 2020 until Monday 4 January 2021. We will not be responding during this time. If you have received a deadline within this time period, please contact us as soon as possible to allow us to extend the deadline. If you receive any automated messages during this time asking you to meet a deadline, we offer apologies and invite you to respond after the festive period or during normal working hours.

With our best for a peaceful festive period and New Year, and we look forward to working with you in 2021.

Dear Dr Liu:

Title: Structure validation of oxidized poly (2-hydroxyethyl acrylate) with multiple aldehyde groups and its application for collagen modification
Manuscript ID: RSOS-201892

Thank you for submitting the above manuscript to Royal Society Open Science. On behalf of the Editors and the Royal Society of Chemistry, I am pleased to inform you that your manuscript will be accepted for publication in Royal Society Open Science subject to minor revision in accordance with the referee suggestions. Please find the reviewers' comments at the end of this email.

The reviewers and handling editors have recommended publication, but also suggest some minor revisions to your manuscript. Therefore, I invite you to respond to the comments and revise your manuscript.

Because the schedule for publication is very tight, it is a condition of publication that you submit the revised version of your manuscript before 01-Jan-2021. Please note that the revision deadline will expire at 00.00am on this date. If you do not think you will be able to meet this date please let me know immediately.

- 1) A text file of the manuscript (tex, txt, rtf, docx or doc), references, tables (including captions) and figure captions. Do not upload a PDF as your "Main Document".
- 2) A separate electronic file of each figure (EPS or print-quality PDF preferred (either format should be produced directly from original creation package), or original software format)
- 3) Included a 100 word media summary of your paper when requested at submission. Please ensure you have entered correct contact details (email, institution and telephone) in your user account

- 4) Included the raw data to support the claims made in your paper. You can either include your data as electronic supplementary material or upload to a repository and include the relevant doi within your manuscript
- 5) All supplementary materials accompanying an accepted article will be treated as in their final form. Note that the Royal Society will neither edit nor typeset supplementary material and it will be hosted as provided. Please ensure that the supplementary material includes the paper details where possible (authors, article title, journal name).

Kind regards,
Dr Laura Smith
Publishing Editor, Journals

On behalf of the Subject Editor Professor Anthony Stace and the Associate Editor Professor Chaohua Cui.

RSC Associate Editor:
Comments to the Author:
(There are no comments.)

RSC Subject Editor:
Comments to the Author:
(There are no comments.)

Reviewer comments to Author:
Reviewer: 1

Comments to the Author(s)
This manuscript is well organized and written. The authors focused on the covalent cross-linking occurred between the aldehyde groups of OP and the amino groups of collagen under alkaline condition, which sounds very interesting. Here I just wonder that could the author give more evidence on the formation of schiff-base, or could the authors point out the feature of this new

chemical bonds via FTIR? Another question, according to my previous study, the concentration of collagen solutions for AFM images in this work were relatively high, I suggest the author do this test at lower concentration.

Reviewer: 2

Comments to the Author(s)

The manuscript presents a kind of oxidized poly (2-hydroxyethyl acrylate) with multiple aldehyde groups for collagen modification. Though the raw materials the authors used lacks of enough novel content, the idea may be interesting for the collagen researcher. However, the manuscript should be rewritten more carefully. It is so hard to understand the manuscript due to irregular writing. The introductions need to pay more attention to the statement on the main innovative points. The analysis should also go deeper. The performances such as mechanical properties, cytotoxicity should be added to make the study more integrated.

Author's Response to Decision Letter for (RSOS-201892.R0)

See Appendix A.

Decision letter (RSOS-201892.R1)

Dear Dr Liu:

Title: Structure validation of oxidized poly (2-hydroxyethyl acrylate) with multiple aldehyde groups and its application for collagen modification
Manuscript ID: RSOS-201892.R1

It is a pleasure to accept your manuscript in its current form for publication in Royal Society Open Science. The chemistry content of Royal Society Open Science is published in collaboration with the Royal Society of Chemistry.

Royal Society of Chemistry
Thomas Graham House

Science Park, Milton Road
Cambridge, CB4 0WF
Royal Society Open Science - Chemistry Editorial Office

On behalf of the Subject Editor Professor Anthony Stace and the Associate Editor Professor
Chaohua Cui.

RSC Associate Editor
Comments to the Author:
(There are no comments.)

Reviewer(s)' Comments to Author:

Appendix A

Dear Editors and Reviewers:

Thank you for your letter and for the reviewers' comments concerning our manuscript entitled "Structure validation of oxidized poly (2-hydroxyethyl acrylate) with multiple aldehyde groups and its application for collagen modification (ID: RSOS-201892).

Those comments are all valuable and very helpful for revising and improving our paper, as well as the important guiding significance to our researches. We have studied comments carefully and have made correction which we hope meet with approval.

Revised portion are marked in red in the paper. The main corrections in the paper and the responds to the reviewer's comments are as flowing:

Responds to the reviewer's comments:

Reviewer #1:

1. Response to comment: (give more evidence on the formation of Schiff-base, or point out the feature of this new chemical bonds via FTIR)

Response:

We have made correction according to the Reviewer's comments. The FTIR spectrum of OP was added into the original spectra (Fig. 7A) to further verify the crosslinking reaction between OP and collagen and the formation of Schiff-base. By comparison, the significant contrasts were formed between the FTIR spectra of OP and OPC. The characteristic absorption peak located at 1730 cm^{-1} was attributed to the carbonyl group (including aldehyde/carboxyl) of OP. Meanwhile, the characteristic absorption peak appeared at $820\text{-}880\text{ cm}^{-1}$ in fingerprint region was generally considered as the hemiacetal and hydrated aldehyde. These information confirmed the structural

characteristic of multiple aldehyde groups of OP. However, after reacted with collagen, the absorption peak at 1730 cm^{-1} disappeared completely, and the absorption intensity of hemiacetal and hydrated aldehyde at $820\text{-}880\text{ cm}^{-1}$ decreased with the increasing reaction pH. These facts confirmed the crosslinking reaction between the aldehyde groups of OP and amino groups of collagen. The imine linkage ($-\text{C}=\text{N}$) formed due to the Schiff's base crosslinking reaction was reported at 1640 cm^{-1} , which appeared in the spectra of OPC when the reaction pH was below 8. When pH was above 8, the peak at 1640 cm^{-1} in OPC might be overlapped by the amide I band of collagen, which was possibly due to the strong covalent binding between OP and collagen under alkaline condition. Therefore, the FTIR spectra (Fig. 7A) of collagen, OP and OPC verified the crosslinking reaction between OP and collagen and the formation of Schiff-base.

4.2.1 FT-IR analysis

line **1**: “Fig. 7A shows the FT-IR spectra of I-collagen, OP modified collagen (OPC)” have been amended by “Fig. 7A shows the FT-IR spectra of I-collagen, **OP** and OP modified collagen (OPC)”

line **5-6**: “Compared to the spectrum of OP, the aldehyde peak at 1730 cm^{-1} was not visible in the FT-IR spectra of OPC” have been amended by “**Compared to the spectrum of OP (Fig. 7A(g)), the characteristic absorption peak of aldehyde group at 1730 cm^{-1} did not exist in the FT-IR spectra of OPC.**”

2. Response to comment: (the concentration of collagen solutions for AFM images in

this work were relatively high)

Response:

In this paper, collagen solution and OP modified collagen solutions were all diluted to 5 mg/L, and the aim of this part was to evaluate the influence of OP dosage on the aggregation morphology of collagen by AFM. The results showed that OP induced the aggregation of collagen molecules, the aggregation degree of which was strengthened with the increasing of OP dosage. The facts further indicated that the crosslinking reaction between the aldehyde groups of OP and the amino groups of collagen occurred. Reviewer suggest doing this test at lower concentration, we think it very reasonable. In this way, the aggregation morphology of collagen might be better observed by AFM. As is known, novel coronavirus is ravaging the world, unfortunately, the porcine type I collagen we purchased from Sigma-Aldrich LLC. (USA) has ran out. The raw material is not available under this situation. So please forgive us for not being able to do this test at lower concentration as you suggested. Please forgive us. Special thanks to you for your good comments and suggestion.

Reviewer #2:

1. Response to comment: (pay more attention to the statement on the main innovative points in introduction part)

Response:

We have made some corrections for the introduction part according to the Reviewer's comments.

Introduction

line **14**: “therefore necessary to undertake chemical modification to improve the properties of collagen-based materials.” have been amended by “**therefore necessary to undertake chemical modification for collagen to meet the application requirements in different fields**”.

line **21**: “so we have sound reason to believe that OP would generate crosslinking reaction with collagen” have been amended by “**so we have sound reason to believe that OP may have the potential to covalently cross-link with collagen to improve its thermal stability**”.

line **25**: “so the structure-property relations of OP still needs to be further investigate” have been amended by “**yet** the structure-property relations of OP still needs to be further investigate”.

line **26**: “developing macromolecular cross-linkers with multiple aldehyde groups and expanding its applications in preparing collagen-based materials” have been amended by “**developing macromolecular cross-linkers with multiple aldehyde groups for collagen.**”

2. Response to comment: (The performances such as mechanical properties, cytotoxicity should be added to make the study more integrated.)

Response:

Dear Reviewer,

Firstly, special thanks to you for your good comments and valuable suggestion. At

present, the research focused on the structure validation of OP with multiple aldehyde groups and its crosslinking modification for collagen. The structural characteristic of multiple aldehyde groups of OP was further verified by HPLC, GPC and hydroxylamine hydrochloride titration, and OP molecule exhibits good electrophilic reactivity by computational simulation. Based on the structural characteristic of OP and the low thermal stability of natural collagen, OP with multiple aldehyde groups was firstly used as a macromolecular cross-linkers for collagen modification. FTIR, CD and AFM analyses confirmed the covalent crosslinking reaction between the multiple aldehyde groups of OP and the amino groups of collagen. VP-DSC and TG analyses demonstrated that OP improved the thermal stability of collagen effectively. The experimental results could basically support the research purpose and solve the problem raised. Reviewer suggest adding mechanical properties, cytotoxicity test, which exactly will be investigated emphatically in our following research. As is known to all, all the world is suffering from novel coronavirus, and what I want to communicate with you in time is about the raw material, the porcine type I collagen we purchased from Sigma-Aldrich LLC. (USA) has ran out. It is not available for the time being under this situation. So please forgive us for not being able to do these tests right now as you suggested. We will focus on these points when condition permitted in our following research. Special thanks to you for your comments and suggestion.